# Comparison of Flash Drought and Traditional Drought on Characteristics and Driving Forces in Xinjiang

**Jing Zhang** [1,2], **Min Zhang** [1,2,*] , **Jialu Yu** [1,2], **Yang Yu** [1,2] and **Ruide Yu** [1,2]

1   Xinjiang Institute of Ecology and Geography, Chinese Academy of Sciences, Urumqi 830011, China; zhangjing21@mails.ucas.ac.cn (J.Z.); yujialu20@mails.ucas.ac.cn (J.Y.); yuyang@ms.xjb.ac.cn (Y.Y.); ruideyu@ms.xjb.ac.cn (R.Y.)
2   University of Chinese Academy of Sciences, Beijing 100049, China
*   Correspondence: zhangmin1206@ms.xjb.ac.cn

**Abstract:** In the context of climate warming, flash drought has become increasingly frequent, posing significant threats to agriculture, ecosystems, and the environment. Xinjiang, located in an arid and semi-arid region, necessitates a thorough investigation into the distinctions between flash drought and traditional drought, as well as an exploration of the driving forces behind both types of drought. In this study, soil moisture data from ERA5-Land were utilized to construct a framework for the identification of flash drought and traditional drought based on historical events. The Geodetector was employed to explore the factors that influence the spatial heterogeneity of these two drought forms. The findings illuminate that (1) in terms of spatial distribution, traditional drought predominated in southern Xinjiang, while flash drought exhibited greater prominence in northern Xinjiang. (2) Regarding changes in trends, both flash drought and traditional drought exhibited an increasing trend in frequency since the 1990s, with the frequency trend of flash drought passing the significance test ($\alpha \leq 0.05$). Additionally, the severity of both flash drought and traditional drought displayed a noteworthy and statistically significant increase within sliding windows ranging from 46 to 62 years. (3) Concerning the driving forces, precipitation emerged as the principal driving force behind both flash drought and traditional drought. Furthermore, human activities exerted a more substantial influence on traditional drought, and the interactions involving human activities had the potential to significantly amplify the explanatory power of the spatial heterogeneity for both drought types. (4) In terms of the drought risk, a notable variation in the risk of flash drought was observed across various ecological zones, with the highest risk occurring in mildly fragile ecological zones. Furthermore, when comparing the results from 1995 to 2019, the flash drought risk exhibited a marked increase in severely fragile ecological zones. This study enriches the understanding of the dynamics of flash drought and traditional drought in Xinjiang and carries important implications for enhancing the precision of drought monitoring and early warning systems.

**Keywords:** flash drought; frequency; Geodetector; severity; traditional drought; Xinjiang

## 1. Introduction

The United Nations Intergovernmental Panel on Climate Change (IPCC) highlights the ongoing warming of the Earth system in the Sixth Assessment Report on Climate Change [1]. As global temperatures continue to rise, it is anticipated that various regions will experience increasingly pronounced and frequent drought events, particularly in the context of agricultural drought and ecological drought [2]. This underscores the urgency of drought as a prominent concern in national climate change mitigation efforts and positions it as a pivotal research area within the international community's response to climate change [3,4].

Traditional drought is typically characterized by slow onset and long duration, leading to its colloquial name 'creeping drought' [5,6]. In the realm of drought characteristics, the

run theory has emerged as a widely accepted and valuable tool for extracting crucial aspects of drought events [7]. These aspects encompass various parameters, including frequency, duration, severity, peak (representing the most severe drought), and intensity [8]. For instance, Jamro et al. (2020) employed 3-month and 12-month timescales of the standardized precipitation evapotranspiration index to examine the evolution of drought duration, severity, peak, and intensity [9]. Additionally, Raposo et al. (2023) proposed two frequently employed methods based on the run theory to characterize drought and investigate their development and recovery stages [10]. In recent years, there has been a growing recognition of the significance of probabilistic drought characterization, particularly in regions with limited water resources [11–13]. Aksoy et al. (2021) introduced the concept of critical drought intensity–duration–frequency curves, which define drought in terms of return period, rather than relying solely on index-based intensity measures [11]. These approaches have been instrumental in examining the complex interrelationships among the severity, intensity, frequency, and duration of drought events.

In contrast with traditional drought, flash drought represents an extreme form of drought event marked by rapid intensification, short duration, and high intensity. It triggers a swift transition from a normal or partially saturated soil condition to severe or even extreme drought within a short timeframe [14–17]. This sudden onset can have adverse impacts on surface and groundwater resources, agroecosystems, and human well-being [18–23]. Researchers worldwide have conducted a series of studies focusing on the definition, monitoring, and underlying mechanisms of flash drought. Various drought indices, such as soil moisture, evapotranspiration stress index, and standardized evapotranspiration stress index, have been used to construct a framework for identifying and characterizing flash drought [24–28]. Furthermore, the occurrence of flash drought is closely tied to meteorological factors, including abnormally high temperatures, markedly reduced precipitation, heightened atmospheric evaporation demand, and declining soil moisture levels [29–33]. As these meteorological conditions deteriorate, flash drought can evolve into more prolonged agricultural drought, endangering vegetation health and depleting soil moisture. Consequently, this presents challenges for effective water resources management [20,34]. Due to the sub-seasonal nature, understanding the driving factors behind flash drought remains intricate and not yet fully understood [35].

Flash drought has become increasingly prevalent across diverse geographical regions worldwide, observed and identified not only in agricultural areas but also in arid regions, high-altitude zones, and forested landscapes [20]. Xinjiang, a region characterized by arid conditions, is geographically divided into northern and southern regions by the imposing Tianshan Mountains. Southern Xinjiang is marked by arid and hot climatic conditions, while northern Xinjiang exhibits comparatively higher humidity levels. Previous research has indicated that flash drought is primarily observed in the mountainous regions of both northern and southern Xinjiang [6,28]. However, comprehensive studies on flash drought in Xinjiang, particularly its distinctions from traditional drought, remain limited. Furthermore, while precipitation, temperature, evapotranspiration, and vapor pressure deficit have been identified as principal driving forces behind flash drought [36], there is a need for more exhaustive investigations into potential additional influencing factors and their intricate interplay.

This paper is dedicated to conducting a comparative analysis of flash drought and traditional drought in Xinjiang. Its objectives encompass the following aspects: (1) establishing a robust identification framework for both flash drought and traditional drought based on ERA5-Land data; (2) conducting a comprehensive examination of the spatial distribution and variation trends of flash drought and traditional drought in Xinjiang from 1960 to 2021, by utilizing the multi-window Mann–Kendall (MK) trend analysis method; (3) quantifying the driving forces behind flash drought and traditional drought, as well as exploring their interactions, by employing the Geodetector. This study aspires to provide a deeper understanding of the distinctions between flash drought and traditional drought within arid and semi-arid regions.

## 2. Data and Methods

### 2.1. Study Area

Xinjiang, located in the northwest region of China, exhibits a temperate continental arid climate, as illustrated in Figure 1. The region boasts a complex and diverse geomorphology, featuring a sequence of distinct geographical formations from north to south, including the Altai Mountains, the Junggar Basin, the Tianshan Mountains, the Tarim Basin, and the Kunlun Mountains. This distinctive topographical arrangement, often referred to as the 'three mountains sandwiching two basins', gives rise to a wide range of climates, vegetation types, and other natural characteristics [37]. The predominant vegetation types in Xinjiang consist of meadows and steppes. In the arid southern regions of Xinjiang, desert landscapes are primarily covered by drought-resistant shrubs, while the northern Tianshan Mountains region is characterized by forested and meadow areas, predominantly featuring evergreen coniferous forests and alpine meadows [38]. The soil types in northern Xinjiang are predominantly composed of Caliche soil and Aridisol. Saline soil is commonly found along the margins of the Tarim Basin, while alpine soil prevails in the mountainous regions. Areas with high Normalized Difference Vegetation Index values are notably located in the Tianshan and Altay mountains of northern Xinjiang, as well as in the peripheral regions of the Tarim Basin in southern Xinjiang. Furthermore, the geographical distance of Xinjiang from the ocean results in limited moisture transport, due to the obstructing influences of the Tibetan Plateau and the Tianshan Mountains, ultimately contributing to scarce precipitation in the region [39]. Over the period from 1960 to 2021, the average annual temperature in Xinjiang ranged from −20 °C to 17 °C, with an average annual precipitation level of 156 mm. Precipitation in northern Xinjiang generally surpasses that in the southern region, and mountainous areas receive greater amounts of precipitation compared to the basin areas [40–42].

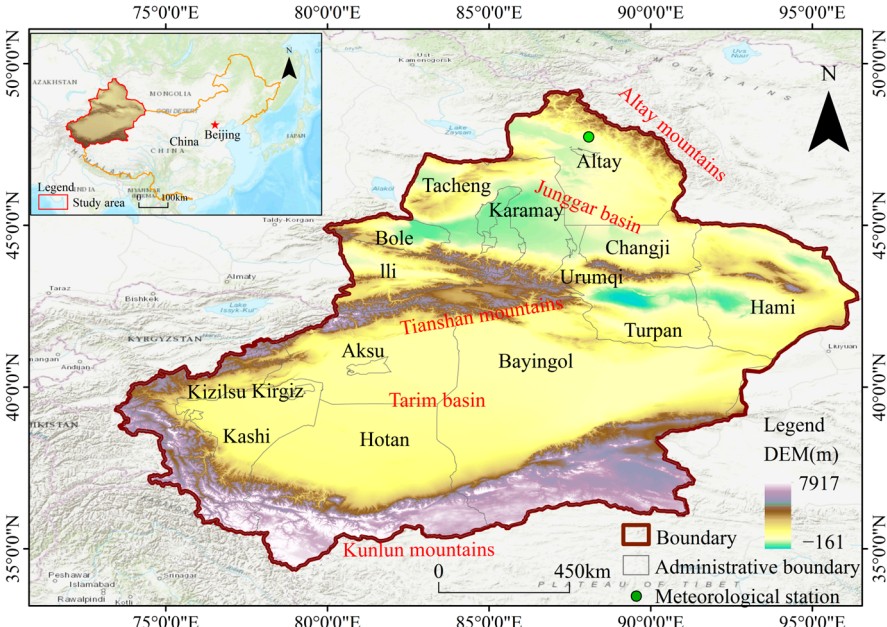

**Figure 1.** Overview of the study area.

### 2.2. Data

Daily meteorological data for Altay spanning the period from 1960 to 2014, encompassing variables such as average temperature (°C), maximum temperature (°C), minimum temperature (°C), precipitation, sunshine hours (h), relative humidity (%), and 2 m wind speed (m/s), were obtained from the meteorological stations. These data sources are part of the daily dataset (V3.0) of essential meteorological elements collected from national ground-based meteorological stations in China, and they are accessible through the Na-

tional Tibetan Plateau Data Center (NTPDC) (https://data.tpdc.ac.cn, accessed on 1 March 2023). Additionally, daily soil moisture for four different soil depths (0–7 mm, 7–28 mm, 28–100 mm, and 100–289 mm) was obtained from ERA5-Land, the fifth generation European Centre for Medium-Range Weather Forecasts (ECMWF) atmospheric reanalysis data of the global climate (https://cds.climate.copernicus.eu, accessed on 1 March 2023) [8,10]. For the purposes of drought identification in this study, a high spatial resolution of 0.1° and soil moisture data from 0 to 100 cm were utilized [9]. Furthermore, various other data from ERA5-Land were employed for drought analysis in this research, covering the years from 1960 to 2021. These datasets include temperature (TEM), total evapotranspiration (ET), wind speed (WIN), precipitation (PRE), and snowmelt (SNM) data (Figure 2).

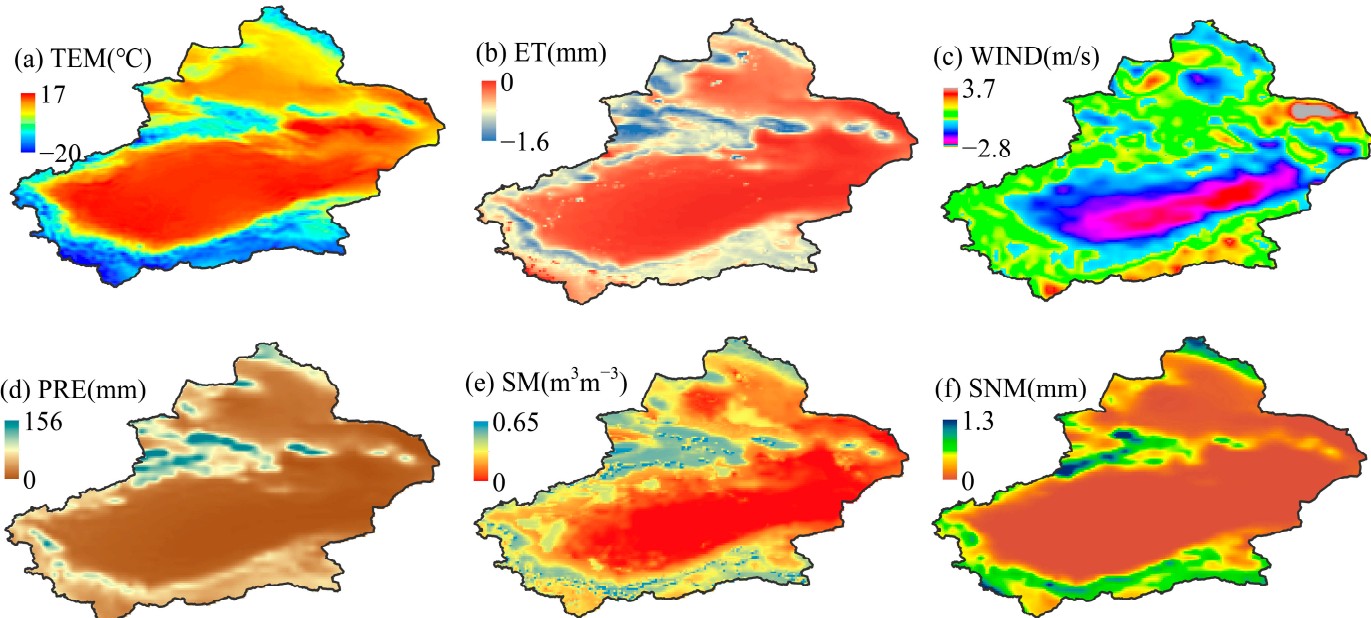

**Figure 2.** The spatial distribution of (**a**) temperature (TEM), (**b**) total evapotranspiration (ET), (**c**) wind speed (WIN), (**d**) precipitation (PRE), (**e**) soil moisture (SM), and (**f**) snowmelt (SNM).

A range of multi-source datasets, as detailed in Table 1, were incorporated into this study. These datasets encompass vegetation type (VEG), soil type (SOIL), topography type (GEO), population density (POP), economic density (GDP) data, and land use (LUCC), all of which were made available by the Resource and Environment Science and Data Center (RESDC) of the Chinese Academy of Sciences (https://www.resdc.cn, accessed on 1 March 2023). Furthermore, normalized difference vegetation index (NDVI) data were sourced from the NTPDC. The spatial distribution of above variables is shown in Figures 3 and 4.

**Table 1.** Data sources and information.

| Data | Date | Data Source | Data Format |
|---|---|---|---|
| VEG | — | RESDC | shp |
| SOIL | — | RESDC | shp |
| GEO | — | RESDC | shp |
| POP (Person/km$^2$) | 1995, 2019 | RESDC | 1 km × 1 km Raster |
| GDP (10$^4$/km$^2$) | 1995, 2019 | RESDC | 1 km × 1 km Raster |
| LUCC | 1995, 2020 | RESDC | shp |
| NDVI | 1982–2015 | NTPDC | 1/12 × 1/12 Raster |

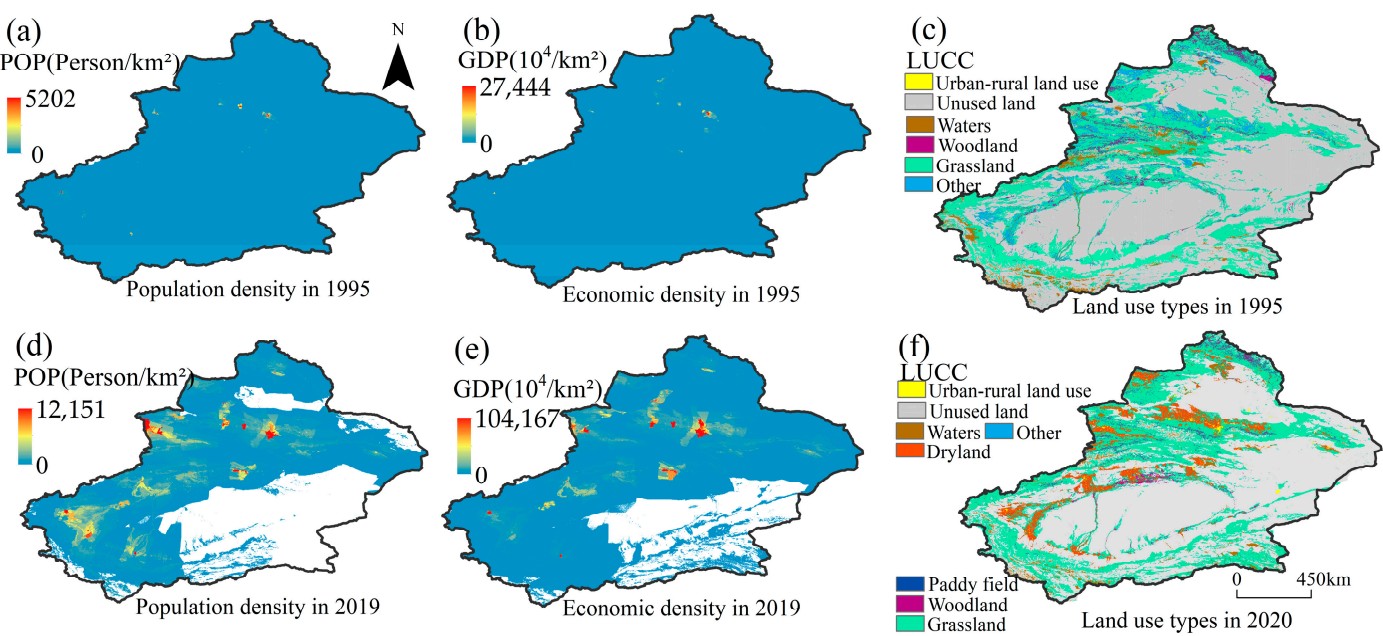

**Figure 3.** Spatial distribution of (**a**,**d**) population density (POP), (**b**,**e**) economic density (GDP), and (**c**,**f**) land use types (LUCC).

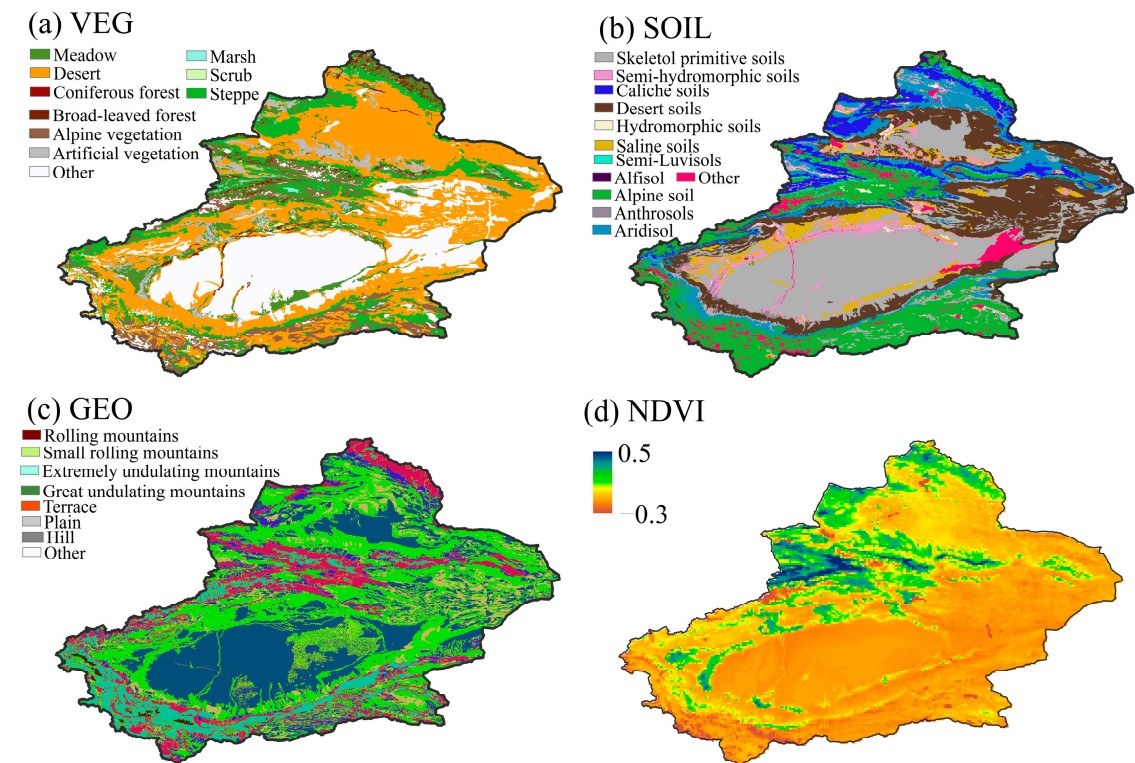

**Figure 4.** The spatial distribution of (**a**) vegetation type (VEG), (**b**) soil type (SOIL), (**c**) topography type (GEO), and (**d**) normalized difference vegetation index (NDVI).

To account for the spatial variability associated with human activities and environmental factors in relation to drought distribution, this study has integrated the ecological zoning (ECZ) scheme as an influential factor. This zoning scheme plays a crucial role in assessing the risk of flash drought and traditional drought across distinct ecological zones,

thereby enhancing the robustness of the analysis [43]. Further details of the ecological zoning scheme can be found in Table 2.

**Table 2.** Ecological zoning scheme.

| Zoning Type | Prefecture-Level City and Prefecture | ED | EE |
|---|---|---|---|
| Stable | Urumqi, Karamay | extremely high | medium |
| Mildly fragile | Ili | medium | extremely high |
| Moderately fragile | Changji, Bayingol, Tacheng, Hami, Bole, Altay, Turpan | relatively high | low |
| Severely fragile | Aksu, Kizilsu Kirgiz, Kashi | relatively low | relatively low |
| Extremely fragile | Hotan | extremely low | extremely low |

ED: the level of economic development; EE: the level of ecological environment.

### 2.3. Methods

2.3.1. Calculation of Hydro-Meteorological Indicators

Flash drought is often triggered by short-term climatic anomalies, leading to the rapid onset [18,36,44–46]. To investigate this phenomenon, anomaly calculations for each hydro-meteorological parameter were conducted using an 8-day time step, where 1 TS represents an 8-day interval. These anomalies were computed using the Z-score method. The calculation of potential evapotranspiration (PET) was executed utilizing the Penman–Monteith formula [47], as described in Equation (1). Furthermore, the vapor pressure deficit (VPD), a critical indicator of air drying [48], was calculated meticulously following the procedures outlined in Equations (2)–(6).

$$\mathrm{PET} = \frac{0.408\Delta(R_n - G) + \gamma\frac{900}{T+273}U_2(e_a - e_d)}{\Delta + \gamma(1 + 0.34U_2)} \tag{1}$$

where PET represents the potential evapotranspiration (mm); $\Delta$ represents the slope of temperature with saturated water vapor pressure (kPa·°C$^{-1}$); U$_2$ represents the wind speed at 2 m above ground (m·s$^{-1}$); $e_a$ is saturated vapor pressure (kPa); $e_d$ is actual vapor pressure (kPa); $T$ is mean air temperature (°C); $\gamma$ is the moisture table constant (kPa·°C$^{-1}$); and $G$ represents soil heat flux density (MJ·m$^{-2}$·d$^{-1}$).

$$VPD = SVP - AVP \tag{2}$$

$$AVP = 6.112 \times f_w \times e^{\frac{17.67T_W}{T_W + 243.5}} \tag{3}$$

$$SVP = 6.112 \times f_w \times e^{\frac{17.67T_a}{T_a + 243.5}} \tag{4}$$

$$f_w = 1 + 7 \times 10^{-4} + 3.46 \times 10^{-6}P_{nst} \tag{5}$$

$$P_{nst} = P_{nsl}\left(\frac{(T_a + 273.16)}{(T_a + 273.16) + 0.0065 \times Z}\right)^{5.625} \tag{6}$$

where $SVP$ and $AVP$ are saturated vapor pressure and actual vapor pressure (kPa), respectively. $T_a$ is the land air temperature (°C). $T_W$ is the dew point temperature (°C). $P_{nst}$ is the air pressure (hPa). $P_{nsl}$ is the air pressure at mean sea level (1013.25 hPa). $Z$ is the altitude (m).

Soil moisture exhibits notable regional variations and seasonal fluctuations, which poses challenges when utilizing it directly for extensive, long-term drought analyses [14]. To address this challenge, this paper chose to divide each year into 46 distinct time intervals, using an 8-day time scale after excluding 29th February during leap years. The last time interval, denoted as 46TS, represented the mean of the last 5 days of each year. The calculation of soil moisture percentiles was executed by fitting an optimal probability

distribution function, as illustrated in Figure 5. Subsequently, these derived probabilities were applied to the original time series and then aggregated to create annual datasets. This consolidated dataset was then employed to identify instances of both flash drought and traditional drought.

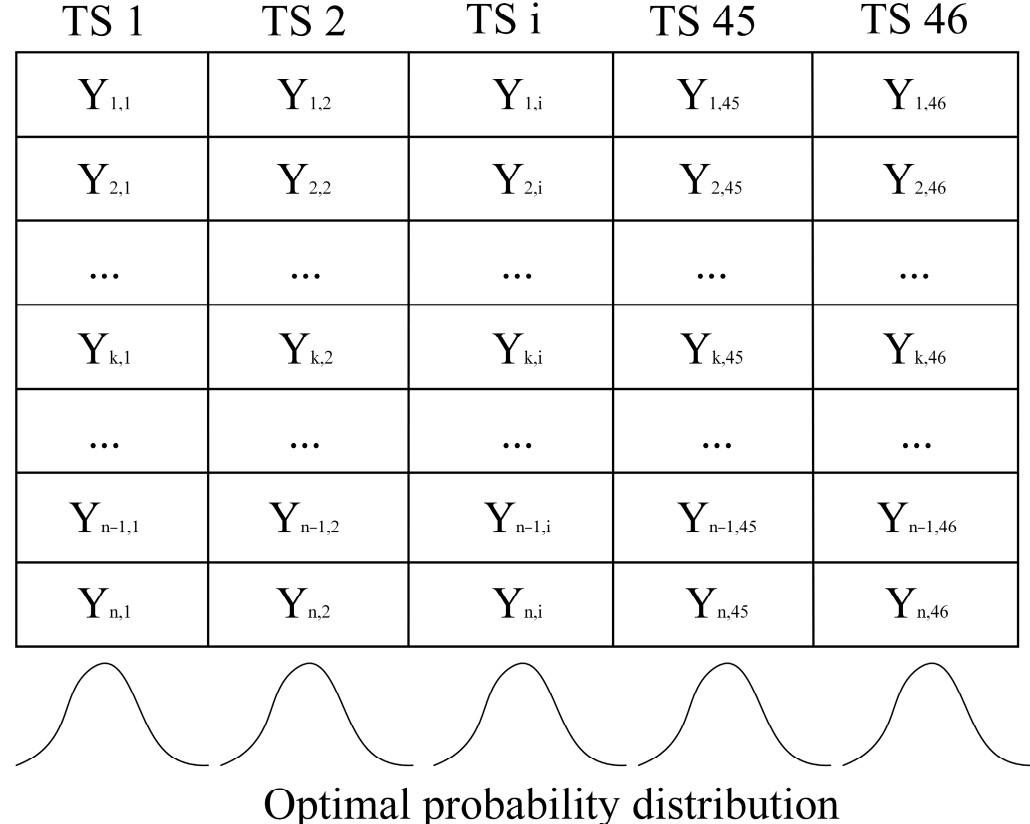

**Figure 5.** Diagram for calculating soil moisture percentile. $Y_{n,i}$ represents soil moisture for the ith TS of the nth year.

### 2.3.2. Identification of Flash Drought and Traditional Drought

The duration of the intensification period served as a criterion to distinguish between flash drought and traditional drought. Specifically, when the intensification period lasted less than 8 TS, it was categorized as a flash drought event; when it exceeded 8 TS, it was classified as a traditional drought event. These distinctions are visually depicted in Figure 6, where Event 1 represents a flash drought event, reducing the soil moisture percentile from 67% to 20% over a span of 40 days (equivalent to 5 TS), while Event 2 portrays a traditional drought event, which took 88 days (equivalent to 10 TS) to transition from normal to dry conditions.

Identifying a drought event involved considering the rapid intensification, the sustenance phase, and the subsequent recovery period [28,35]. The drought identification part of Figure 7 outlines the process for both flash drought and traditional drought, encompassing three criteria for delineating the intensification, sustenance, and recovery phases: (1) The onset of the drought was marked by the moment before the soil moisture percentile descended below 40%. This ensures that the soil moisture transitions from a normal or wet state to a dry state. Likewise, the termination of the intensification period was denoted when the soil moisture percentile fell below 20%, indicating the onset of an actual drought condition. The phase lasting less than 8 TS was deemed the onset of a flash drought event, and conversely, it marked the commencement of a traditional drought; (2) The sustenance phase was characterized by a soil moisture percentile below 20%, remaining relatively low without a discernible upward trend; (3) The recovery or end of the drought was signaled

by the soil moisture percentile once again surpassing 20% and exhibiting a clear upward trend. When it exceeded 40%, it indicated the end of the drought.

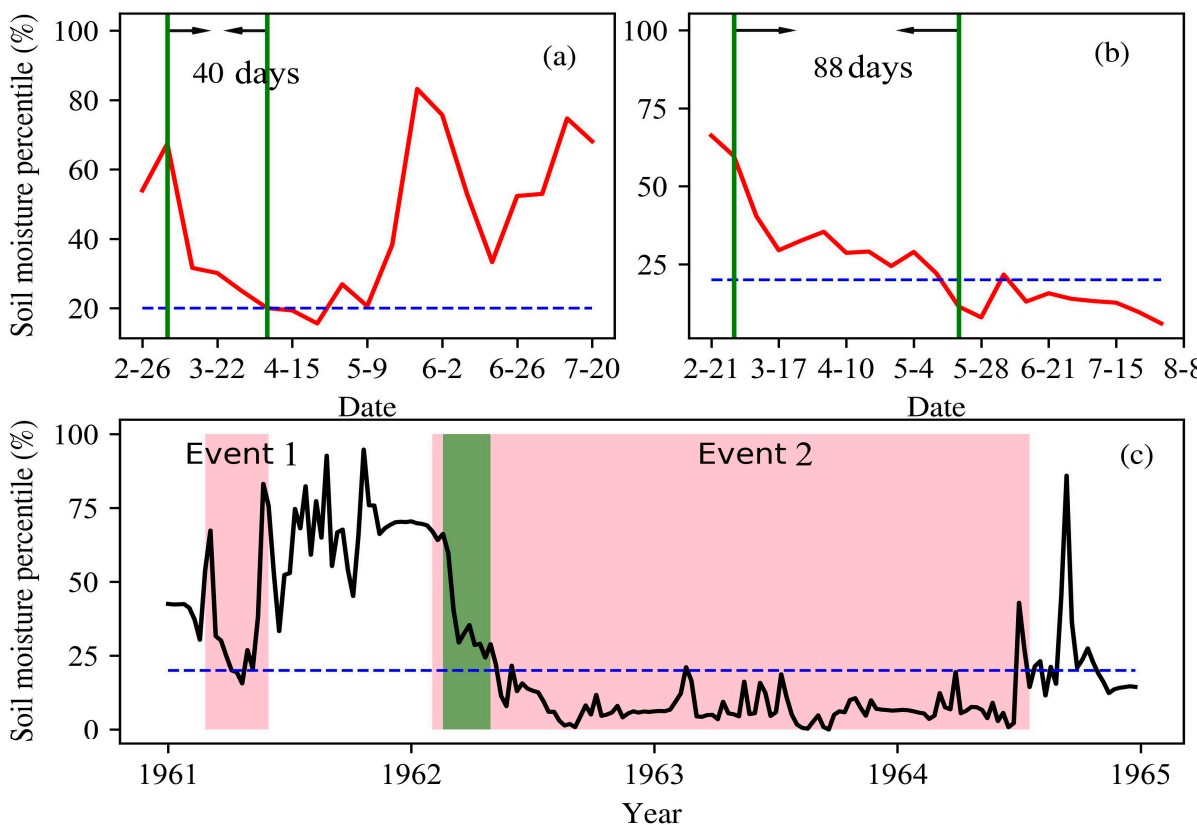

**Figure 6.** Flash drought and traditional drought in Altay from 1961 to 1965 (shown in (**c**)). The intensification periods of Event 1 and Event 2 are shown between the green lines of (**a**,**b**).

The run theory was employed to identify both flash drought and traditional drought within the time series for each grid, adhering to the criteria outlined previously [36]. The characterization of these drought events encompassed four key metrics: frequency, duration, intensity, and severity. Frequency denoted the count of all drought events identified within the time series for each grid. Duration quantified the number of time steps between the commencement and the end of a drought event [10]. Mean duration was calculated as the average duration across all identified drought events within the grid. Severity measured the cumulative soil moisture anomaly during the duration of a drought event [9]. In addition to the above, two additional metrics related to drought intensity were computed. Intensity was defined as the severity of a drought event divided by its duration [10], and average intensity represented the mean intensity across all identified drought events within the grid.

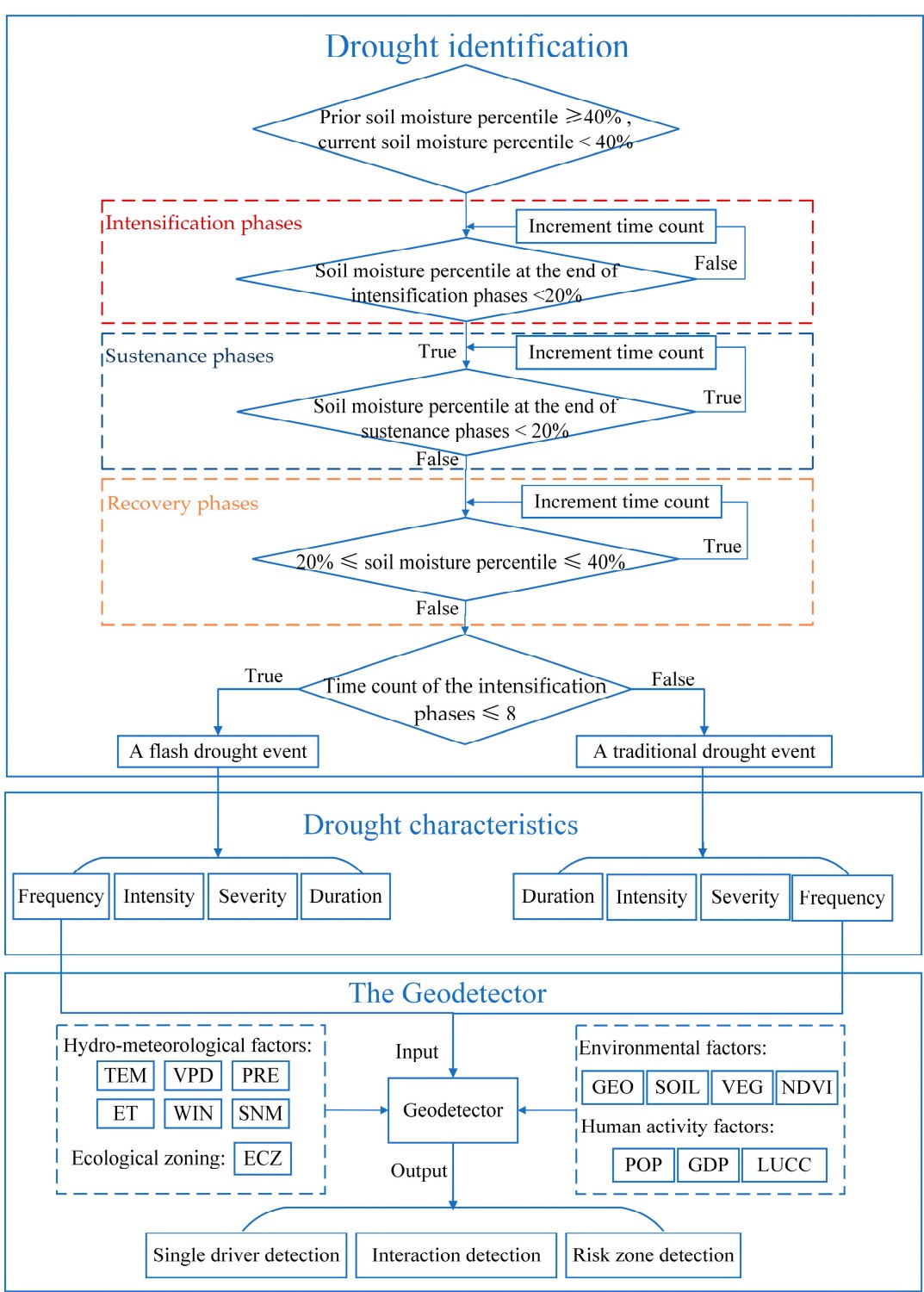

**Figure 7.** A flowchart of the comprehensive methodology employed in this study.

### 2.3.3. The Geodetector

The Geodetector as a statistical methodology was used to detect spatial variations and unveil the factors influencing flash drought and traditional drought [49,50]. The Q value served as a metric to gauge the extent of influence exerted by hydro-meteorological

variables, environmental factors, and human activities on both flash drought and traditional drought. It was expressed as follows:

$$Q = 1 - \frac{\sum_{i=1}^{n} N_i \alpha_i^2}{N\alpha^2},  \tag{7}$$

where i = 1, . . ., n is the stratification of impact factors, $N_i$ and $N$ are the number of cells in stratum *i* and the whole area, and $\alpha_i^2$ and $\alpha^2$ are the variance of stratum *i* and the whole area. The range of Q values is [0, 1], where Q = 0 means that the drought event is independent of the variables. Larger Q values indicate that the variables possess a more pronounced and significant explanatory influence on the drought occurrence. Conversely, smaller Q values suggest a relatively weaker explanatory power for the variables.

The remaining modules include interaction detection, ecological detection, and risk zone detection. In this study, the interaction detection module calculated the interaction between pairs of factors to ascertain whether the combined influence of these factors enhances or diminishes their explanatory capacity concerning flash drought and traditional drought. Meanwhile, the risk zone detection module computed risk values associated with flash drought and traditional drought within distinct ecological zones. The selected variables for analysis encompass hydro-meteorological factors such as TEM, PRE, ET, WIN, VPD, and SNM, human activity factors including POP, GDP, and LUCC, and environmental factors represented by VEG, SOIL, GEO, and NDVI. Additionally, the abovementioned factors together with ECZ served as independent variables, while the frequency of flash drought and traditional drought served as the two dependent variables for exploring spatial variations, as illustrated in Figure 7.

## 3. Results

### 3.1. Analysis of Flash Drought and Traditional Drought in Altay

As illustrated in Figure 6, Altay experienced a flash drought event from March to May in 1961 and a traditional drought event from 1962 to 1965. This subsection is devoted to a comparative analysis of these two events by examining the variations in temperature, potential evapotranspiration, precipitation, and soil moisture in Altay during the period from 1961 to 1962 (refer to Figure 8).

Figure 8 reveals that in March and April of 1961, temperature exceeded the 1960–2014 average by 2.88 °C and 3.56 °C, respectively. This abnormal temperature increase resulted in significantly elevated potential evapotranspiration values compared to the historical average. The conjunction of these factors, combined with the scarcity of precipitation during the winter and spring, triggered a rapid decline in soil moisture starting from early March and persisting until May. This decline indicates that the unusual reduction in precipitation, along with the abnormal increase in temperature and potential evapotranspiration, constituted the primary factors contributing to the flash drought event. In addition, the abnormal increase in precipitation interrupted flash drought development, which is consistent with the finding of Otkin et al. (2019) [51]. However, soil moisture deficits escalated significantly due to the exceptionally warm temperatures observed in the winter of 1961 and the spring of 1962. Despite the recovery of precipitation levels in November, the increased precipitation could not effectively compensate for the soil moisture deficit caused by the severe drought conditions. Consequently, this led to the onset of a traditional drought event that persisted until the end of the year.

The Hovmoller diagram, illustrating climate element anomalies in Altay (45–49°N, 80–90°E), is presented in Figure 9. In this diagram, the vertical axis corresponds to the months, while the horizontal axis represents the longitude. Notably, both events were associated not only with air temperature, precipitation, and potential evapotranspiration, but also with snowmelt. At the onset of the flash drought event in 1961, there was an abnormal increase of 2.5 mm in snowmelt, leading to a recharge of soil moisture, as depicted in Figure 5. Simultaneously, an abnormal rise in air temperature (temperature anomaly >1 °C) resulted in arid atmospheric conditions (vapor pressure deficit anomaly

>1.5 kPa). Additionally, wind speed also exhibited an abnormal increase. The combined impact of these factors prompted a relatively rapid transition of soil moisture from a wet state to a deficit state. However, during the spring of 1962, the absence of precipitation and the limited recharge from snowmelt caused soil moisture to hover slightly below the multi-year average. The transition of soil moisture into a deficit state occurred more gradually, driven by warming temperatures and elevated potential evapotranspiration during the spring. Subsequently, the severe soil moisture deficit during the summer and autumn (soil moisture anomaly $<-1$ m$^3$m$^{-3}$) was exacerbated by high-temperature anomalies, increased potential evapotranspiration anomalies, and reduced precipitation levels during the summer.

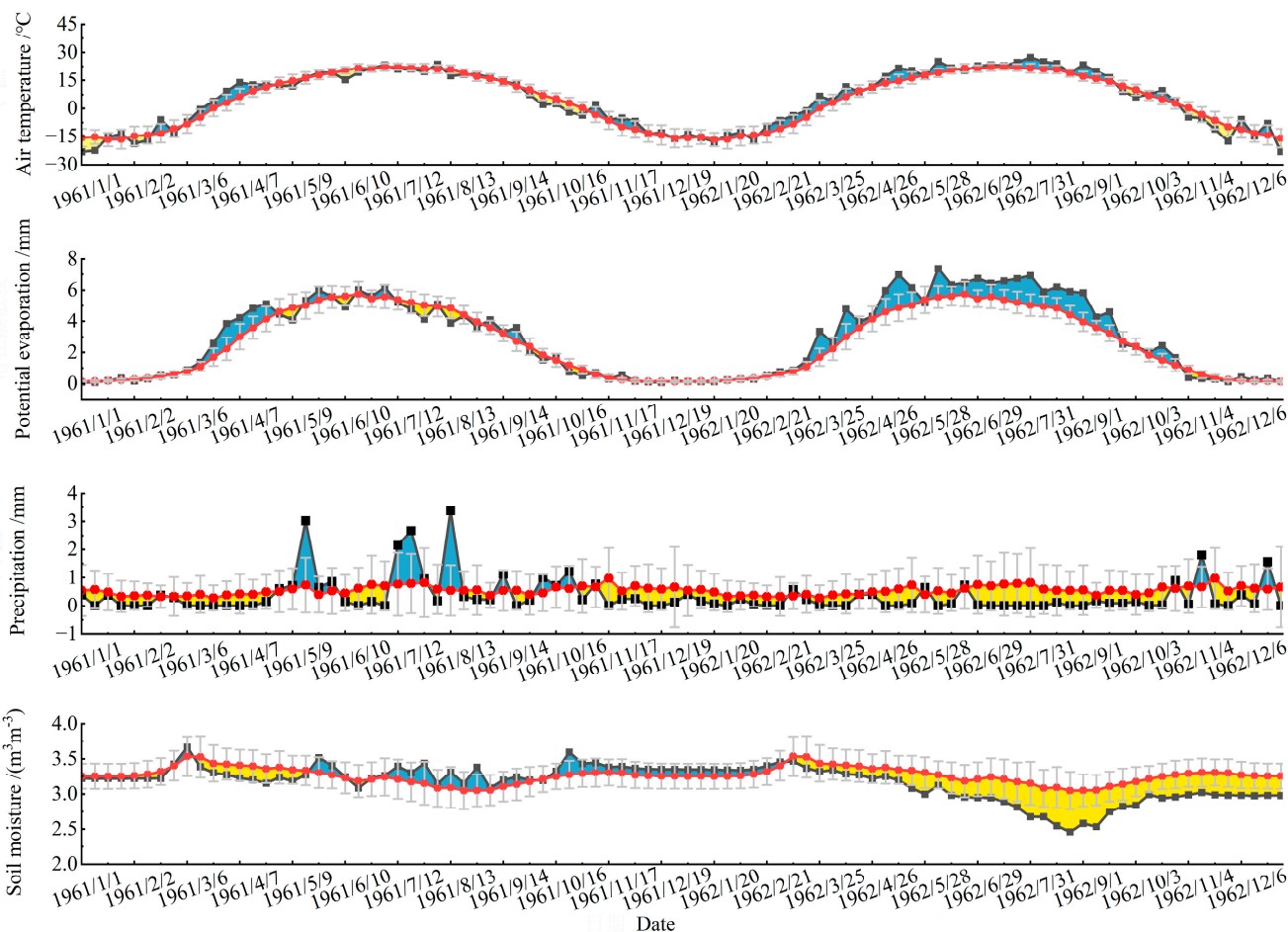

**Figure 8.** Temporal series of 1-TS in air temperature, potential evapotranspiration, precipitation, and soil moisture in Altay (47.73°N, 88.08°E). The black line with markers represents 1961–1962, and the red line denotes the 1960–2014 average. The gray vertical bars indicate mean ± standard deviation. The blue shading signifies values exceeding the mean, while the yellow shading indicates values below the mean.

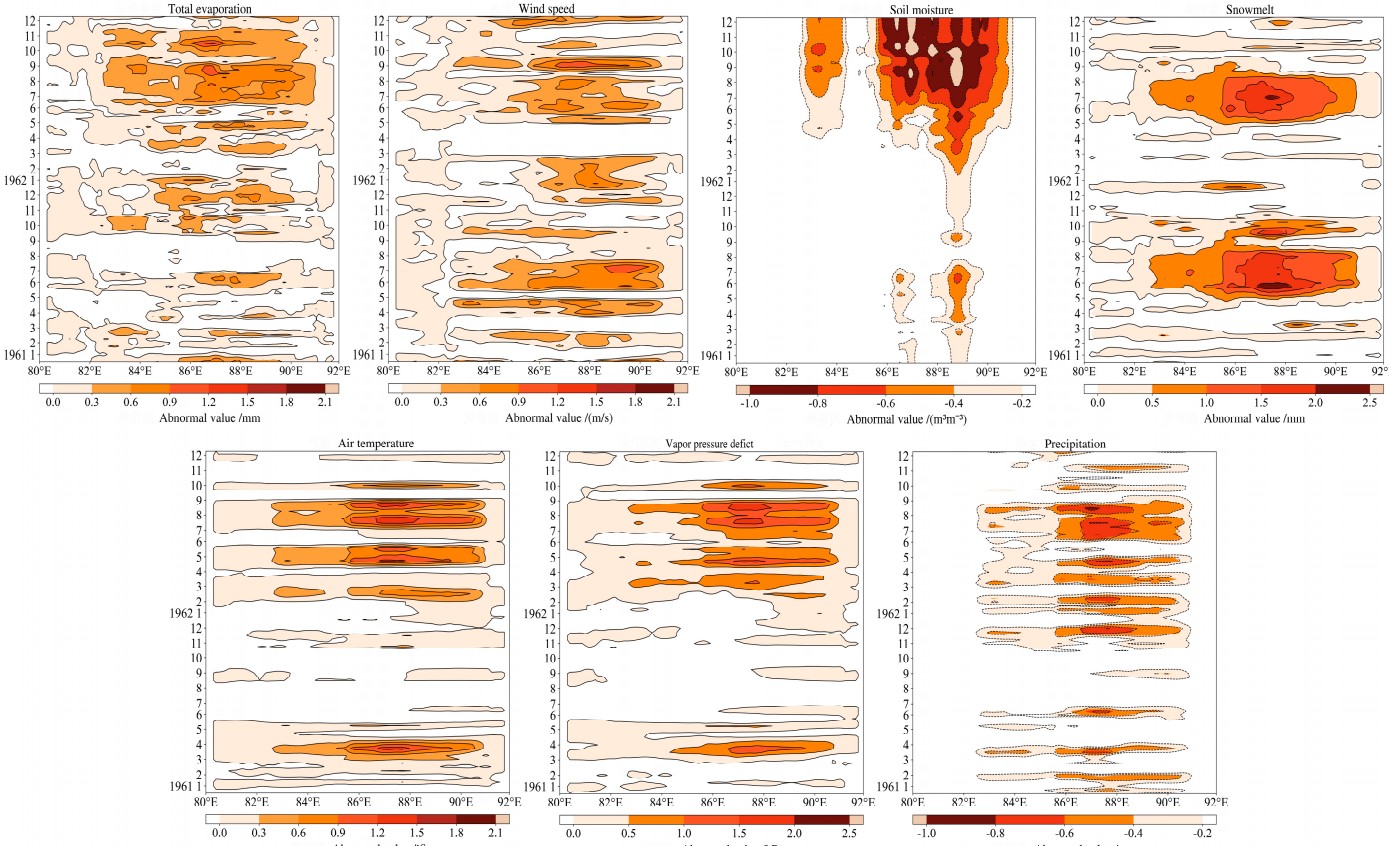

**Figure 9.** Hovmoller diagram of abnormal climatic elements (total evaporation, wind speed, soil moisture, snowmelt, air temperature, vapor pressure deficit, precipitation) in Altay (45–49°N, 80–90°E) from 1961 to 1962.

### 3.2. Spatial Characterization

The spatial distribution of frequency, duration, intensity, and the proportion of flash drought and traditional drought in Xinjiang for the period of 1960–2021 is depicted in Figure 10, revealing a distinct pattern within the region. In northern Xinjiang, flash drought predominated, accounting for over 80% of drought events, with a frequency ranging from five to ten occurrences. Conversely, in southern Xinjiang, particularly in the periphery of the Tarim Basin, traditional drought held a higher proportion, exceeding 80%, and occurred at a frequency of one to five times. In addition, traditional drought tended to be more prolonged, often persisting for over 30 TS and sometimes extending up to 40 TS. In southern Xinjiang, drought events exceeding 40 TS were more common, whereas in northern Xinjiang, droughts usually lasted for less than 20 TS.

Figure 11 shows the disparities in the seasonal distribution of frequency and intensity of flash drought and traditional drought across four distinct seasons: spring (March–May), summer (June–August), autumn (September–November), and winter (December–February). At the seasonal scale, differences in intensity between the two types of drought were not notably significant. However, marked disparities in their frequency across different seasons were observed. Flash drought events were predominantly noted in the spring and summer. In northern Xinjiang, the occurrence of flash drought was more frequent, ranging from five to ten instances during the spring and summer but dropping to relatively lower levels (0–5 times) during the autumn and winter. In contrast, during the summer, traditional drought exhibited a higher frequency in southern Xinjiang compared to northern Xinjiang, with the situation reversing in the autumn.

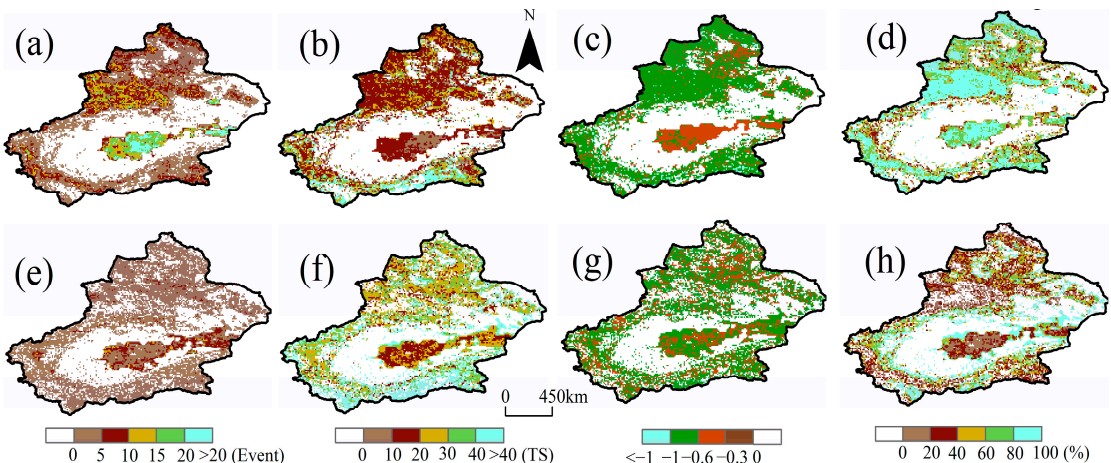

**Figure 10.** Spatial distribution for the frequency, duration, intensity, and proportions of flash drought (**a**–**d**) and traditional drought (**e**–**h**) in Xinjiang from 1960 to 2021. White areas indicate unidentified drought events.

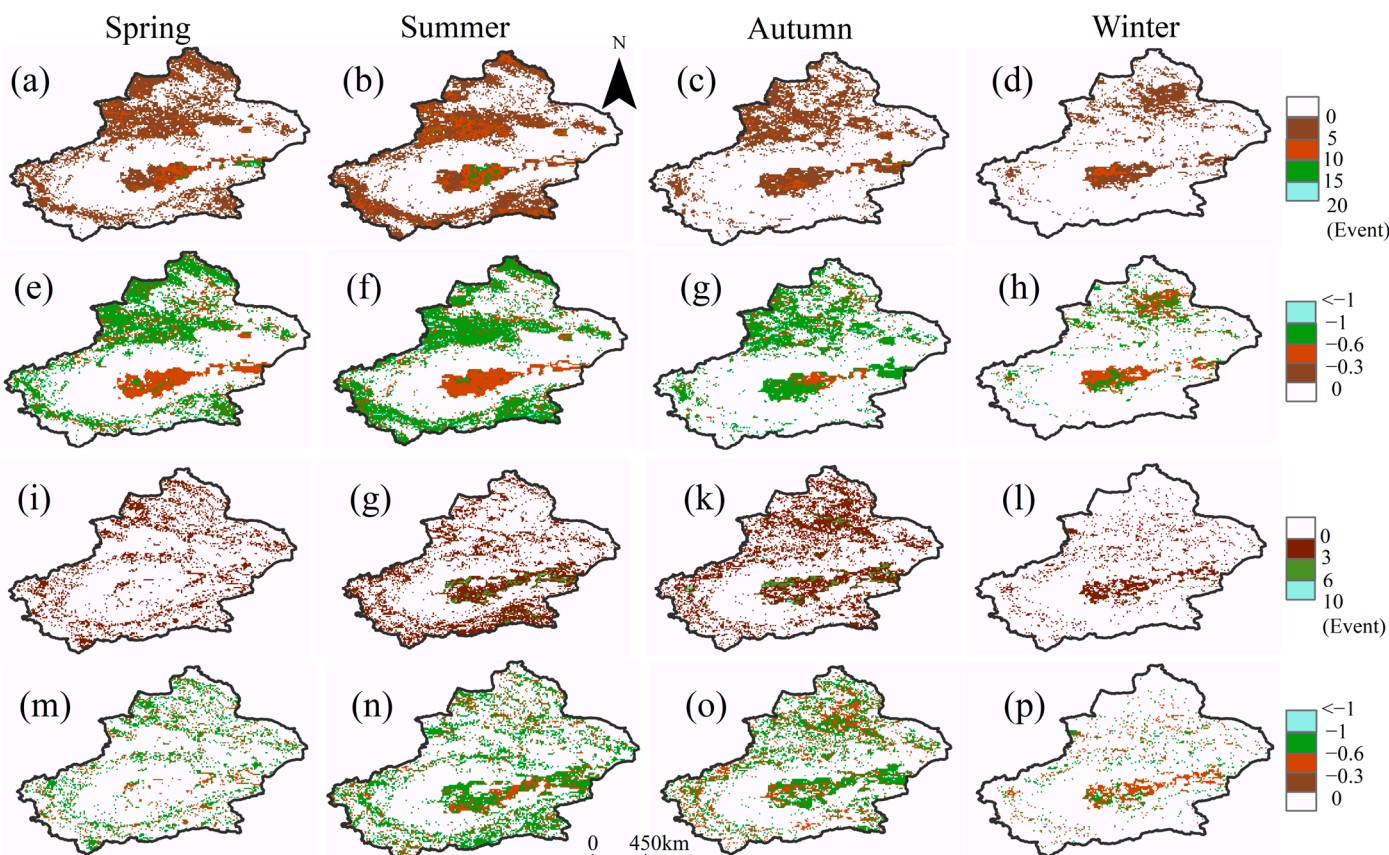

**Figure 11.** Seasonal distribution for the frequency and intensity of flash drought (**a**–**h**) and traditional drought (**i**–**p**) in Xinjiang from 1960 to 2021. White areas indicate unidentified drought events.

To further analyse the variation trends associated with flash drought and traditional drought, a multi-window sliding method was adopted to examine frequency and severity trends spanning from 1960 to 2021, as illustrated in Figure 12. Within the context of sliding windows ranging from six to 16 years, the frequency of flash drought demonstrated a pattern of alternating decreases and increases during the periods 1960–1970, 1970–1980, 1980–1990, and 2000–2021. Conversely, the frequency of traditional drought exhibited

an increasing-decreasing-increasing trend during the periods 1960–1980, 1980–2000, and 2000–2021. As the window size expanded, both flash drought and traditional drought frequencies shifted from a decreasing trend to an increasing trend beginning around the year 2000. Moreover, the increasing trend in the frequency of flash drought passed the significance test ($\alpha \leq 0.05$). Simultaneously, when the window size ranged from 46 to 62 years, the severity of both flash drought and traditional drought displayed a noteworthy and statistically significant increasing trend, also passing the significance test ($\alpha \leq 0.05$). This trend aligned with the historical records of severe drought events [52]. It is important to note that from the 1990s onwards, the frequency of both flash drought and traditional drought exhibited a rising trend, while severity trended downward. This divergence may be associated with recent short-term climate variations marked by warming and increased humidity.

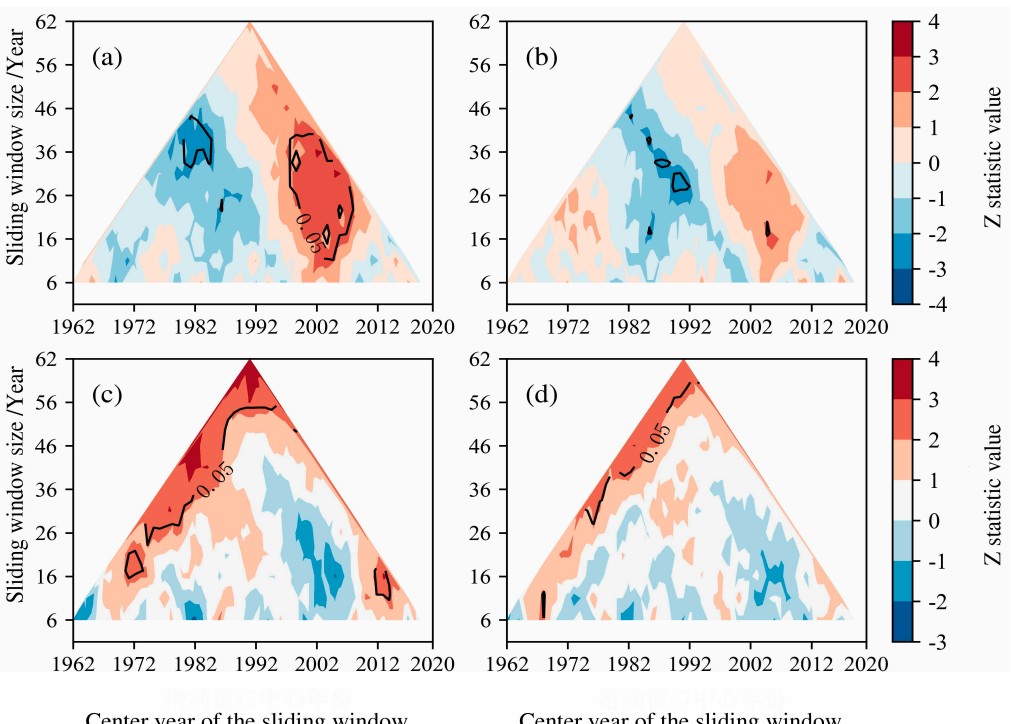

**Figure 12.** Multi-window sliding trend for the frequency and severity of flash drought (**a,c**) and traditional drought (**b,d**) in Xinjiang from 1960 to 2021. When Z is less than 0, it indicates a downward trend, whereas Z greater than 0 signifies an upward trend. Contours indicate passing the 95% confidence interval ($\alpha = 0.05$).

### *3.3. Analysis of the Driving Forces*

#### 3.3.1. Single-Factor Detection and Two-Factor Interaction Detection

The single-factor detection module employed Q values to assess the degree to which individual drivers account for the spatial distribution of flash drought and traditional drought, as depicted in Figure 13. Our analysis focused on anthropogenic data for the years 1995 and 2019 to investigate their respective influences. Notably, the results showed no significant variations between these two years. Therefore, the detection results were present solely for the year 1995, with specific comparative findings between 1995 and 2019 detailed in the subsequent subsection.

The findings reveal that precipitation exerted the most pronounced influence on the spatial variation of both flash drought (Q value: 0.343) and traditional drought (Q value: 0.15). Importantly, all factors exhibited a greater explanatory power for flash drought compared to traditional drought. Further analysis involved calculating the contribution values of the three variable types, revealing that meteorological variables had the highest contribution to both flash drought and traditional drought, accounting for 69.3% and 72.4%

of the respective variations. Environmental variables exhibited a more substantial impact on flash drought (environmental variables: 21.9% > human activities: 8.8%), whereas human activities exerted a more significant influence on traditional drought (human activities: 16.4% > environmental variables: 11.2%). It is worth noting that although the contribution of human activities to flash drought may be relatively modest, its contribution to traditional drought should not be underestimated.

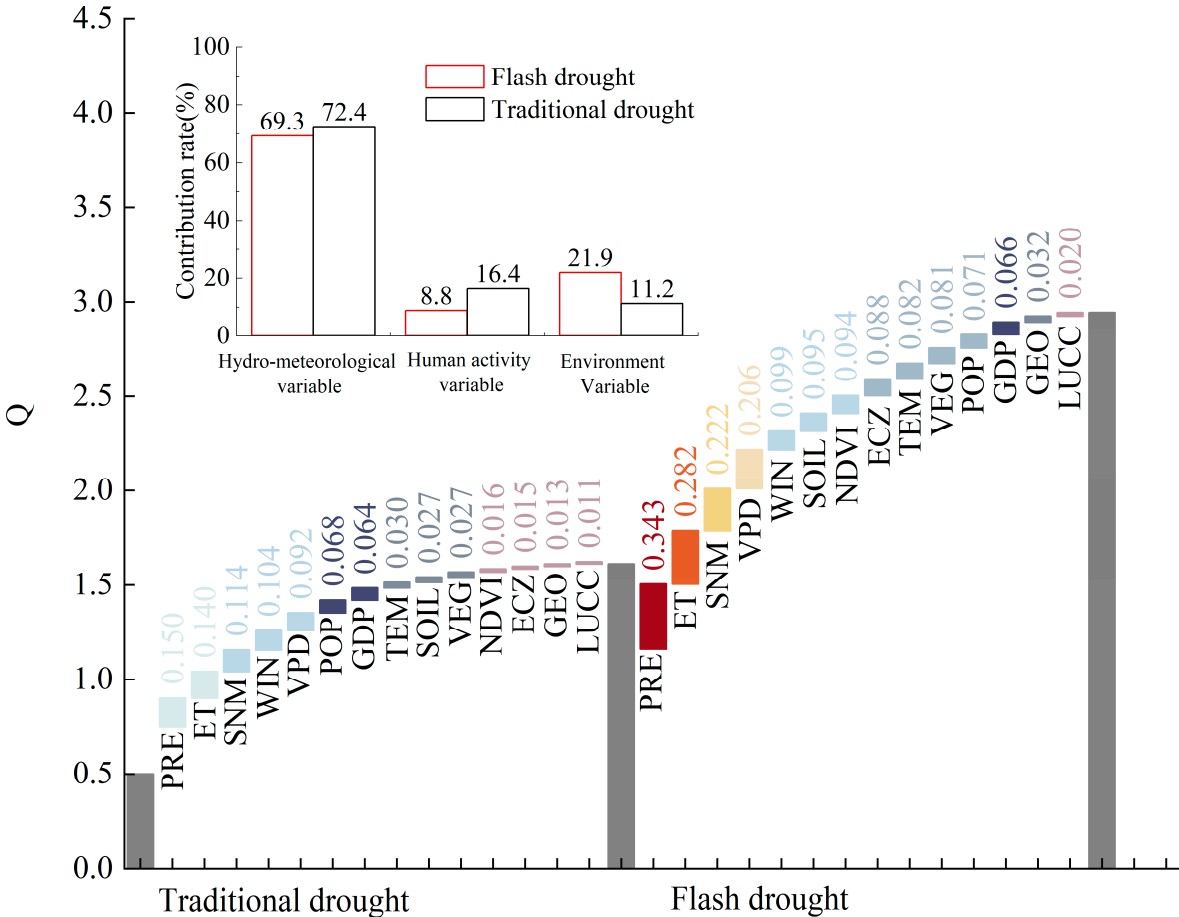

**Figure 13.** Single-factor detection of flash drought and traditional drought.

Tables 3 and 4 quantify the interactions between pairs of the driving factors. The results notably demonstrate that the explanatory power of these interactions exhibited a pattern of two-factor enhancement. This finding signifies that interactions between pairs of factors exerted a more substantial influence on spatial heterogeneity compared to the individual contributions of each factor. Remarkably, the interactions involving human activity variables tended to display predominantly nonlinear enhancement relationships with other factors. Conversely, interactions among meteorological variables tended to exhibit more pronounced two-factor enhancement relationships. This suggests that interactions involving human activities had the potential to significantly amplify the explanatory power of spatial heterogeneity in both flash drought and traditional drought. The influence of human activities on surface environments played a pivotal role in altering and regulating environmental vectors and microclimates [53].

**Table 3.** Two-factor interaction detection of flash drought.

| | ECZ | ET | PRE | SNM | TEM | VPD | WIN | NDVI | GEO | LUCC | SOIL | VEG | POP |
|---|---|---|---|---|---|---|---|---|---|---|---|---|---|
| ET | 0.3221 | | | | | | | | | | | | |
| PRE | 0.3718 | 0.3621 | | | | | | | | | | | |
| SNM | 0.3299 | 0.3697 | 0.4091 | | | | | | | | | | |
| TEM | **0.211 *** | 0.3221 | 0.3814 | 0.2815 | | | | | | | | | |
| VPD | 0.2881 | 0.3317 | 0.3838 | 0.2973 | 0.2706 | | | | | | | | |
| WIN | **0.2129 *** | 0.3373 | 0.3901 | 0.3135 | **0.2242 *** | 0.2630 | | | | | | | |
| NDVI | 0.1579 | 0.3739 | 0.4187 | **0.3877 *** | **0.2618 *** | **0.3492 *** | **0.2589 *** | | | | | | |
| GEO | 0.1298 | 0.3064 | 0.3657 | 0.2424 | 0.1112 | 0.2302 | **0.1645 *** | **0.1365 *** | | | | | |
| LUCC | 0.1031 | **0.3161 *** | **0.3710 *** | 0.2714 * | **0.1522 *** | **0.2576 *** | **0.1640 *** | **0.1198 *** | **0.0599 *** | | | | |
| SOIL | **0.1912 *** | 0.3583 | 0.4118 | 0.2687 | **0.1706 *** | 0.2582 | **0.2494 *** | **0.2240 *** | 0.1174 | **0.1331 *** | | | |
| VEG | **0.1852 *** | 0.3644 | **0.4071 *** | 0.2948 | **0.1963 *** | **0.2846 *** | **0.2100 *** | **0.1972 *** | **0.1269 *** | **0.1015 *** | 0.1715 | | |
| POP | **0.1683 *** | **0.3756 *** | **0.4261 *** | **0.4407 *** | **0.3621 *** | **0.3943 *** | **0.2122 *** | **0.2082 *** | **0.1779 *** | **0.1221 *** | **0.3088 *** | **0.2404 *** | |
| GDP | **0.1596 *** | **0.3746 *** | **0.4271 *** | **0.4448 *** | **0.3614 *** | **0.4017 *** | **0.2050 *** | **0.2032 *** | **0.1735 *** | **0.1188 *** | **0.3093 *** | **0.2355 *** | 0.0816 |

\* The bolded part of the table indicates the two-factor nonlinear enhancement relationship and the rest is the two-factor enhancement relationship.

**Table 4.** Two-factor interaction detection of traditional drought.

| | ECZ | ET | PRE | SNM | TEM | VPD | WIN | NDVI | GEO | LUCC | SOIL | VEG | POP |
|---|---|---|---|---|---|---|---|---|---|---|---|---|---|
| ET | 0.1532 | | | | | | | | | | | | |
| PRE | **0.1773 *** | 0.1681 | | | | | | | | | | | |
| SNM | **0.1595 *** | 0.189 | 0.1881 | | | | | | | | | | |
| TEM | **0.0906 *** | **0.1821 *** | **0.1847 *** | 0.1375 | | | | | | | | | |
| VPD | **0.1393 *** | **0.1816 *** | 0.1803 | 0.142 | 0.1194 | | | | | | | | |
| WIN | **0.148 *** | 0.205 | 0.2176 | 0.1908 | **0.1509 *** | 0.1575 | | | | | | | |
| NDVI | **0.0406 *** | **0.1748 *** | **0.1937 *** | **0.1596 *** | **0.0893 *** | **0.1362 *** | **0.130 *** | | | | | | |
| GEO | **0.0373 *** | **0.1602 *** | **0.1639 *** | **0.1232 *** | 0.0425 | 0.0984 | **0.118 *** | **0.0346 *** | | | | | |
| LUCC | **0.0271 *** | **0.1565 *** | **0.1679 *** | **0.1286 *** | **0.0595 *** | **0.1105 *** | **0.1178 *** | **0.0278 *** | **0.022 *** | | | | |
| SOIL | **0.0827 *** | **0.187 *** | **0.196 *** | **0.1581 *** | **0.0772 *** | **0.1249 *** | **0.1399 *** | **0.0642 *** | **0.0485 *** | **0.0451 *** | | | |
| VEG | **0.0657 *** | **0.177 *** | **0.1851 *** | 0.1385 | **0.0702 *** | 0.1185 | **0.131 *** | **0.0581 *** | 0.0379 | 0.0363 | 0.0516 | | |
| POP | **0.0899 *** | **0.2085 *** | **0.2233 *** | **0.2430 *** | **0.2097 *** | **0.2193 *** | **0.1758 *** | **0.1053 *** | **0.1025 *** | **0.0793 *** | 0.1355 | **0.1475 *** | |
| GDP | **0.0794 *** | 0.2039 | **0.2201 *** | **0.2358 *** | **0.1929 *** | **0.2052 *** | 0.1585 | **0.0988 *** | **0.0965 *** | **0.0751 *** | **0.1234 *** | **0.1322 *** | 0.0731 |

\* The bolded part of the table indicates the two-factor nonlinear enhancement relationship and the rest is the two-factor enhancement relationship.

### 3.3.2. Comparison of the Results of 1995 and 2019

The results for the years 1995 and 2019 are illustrated in Figures 14 and 15, respectively. In 2019, there was a slight increase in the contribution of meteorological variables to both flash drought and traditional drought when compared to the year 1995. In contrast, the contribution of human activities exhibited a slight decrease in 2019 compared with 1995. This shift may reflect changing patterns of human influence on drought occurrences over recent years. Furthermore, it is noteworthy that the contribution of human activities to traditional drought was higher than that for flash drought, possibly indicative of the intensified human impact on traditional drought events, particularly in more recent times.

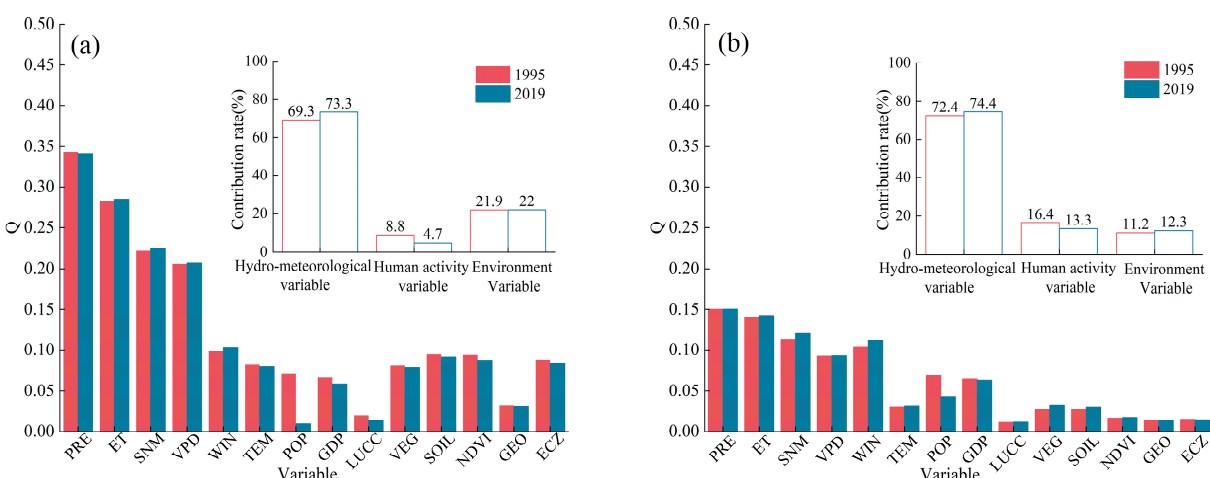

**Figure 14.** Q value of flash drought (**a**) and traditional drought (**b**).

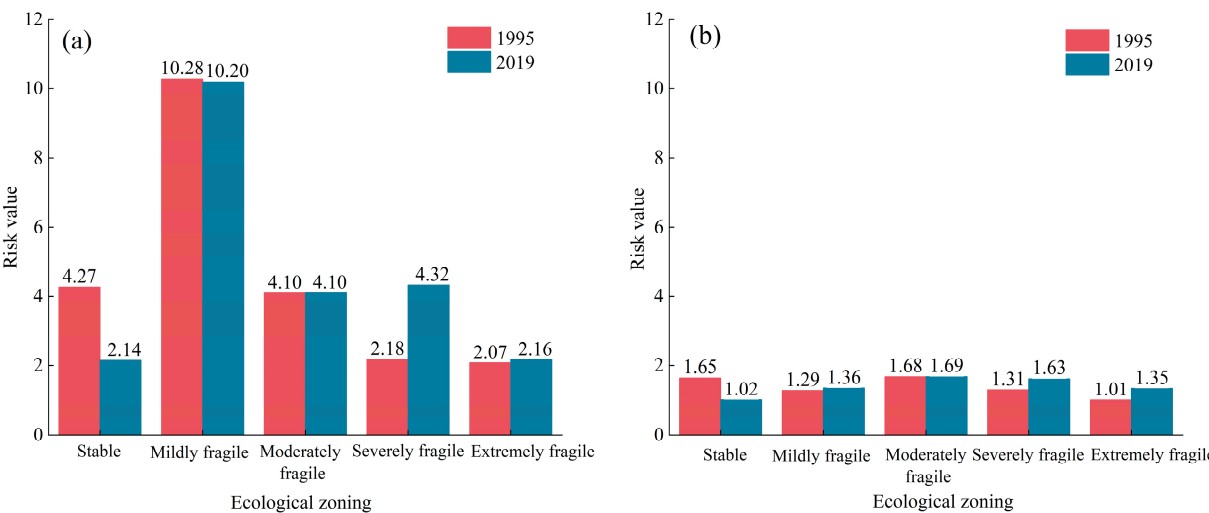

**Figure 15.** Risk value of ecological zoning associated with flash drought (**a**) and traditional drought (**b**).

Additionally, the risk values associated with flash drought and traditional drought were assessed. The analysis uncovered that there was considerable variation in the risk of flash drought occurrence across various ecological zones. The risk of flash drought was highest in the mildly fragile ecological zones (Ili region). In contrast, the risk of traditional drought demonstrated less variability across ecological regions. Furthermore, when comparing the results from 1995 to 2019, the risk of flash drought decreased in stable ecological zones but increased notably in severely fragile ecological zones. These findings underscore the need for heightened attention to flash drought events in Xinjiang in recent years, particularly within the mildly fragile ecological zones.

## 4. Discussion

Over the past six decades, northern Xinjiang has experienced a growing prevalence of flash drought, while southern Xinjiang has witnessed a higher incidence of traditional drought. These trends are consistent with prior research findings [6,28]. The principal factor influencing the spatial distribution of flash drought is precipitation. As depicted in Figure 10d, regions such as the northern Altai Mountains, the central Tianshan Mountains, and the southern Kunlun Mountains exhibit flash drought rates as high as 80%. This can be attributed to significant precipitation gradients and rapid snowmelt in these areas. Under abnormal meteorological conditions, such as sharp declines in precipitation, soil moisture can rapidly decrease, triggering the onset of flash drought. In the context of global warming, the increased frequency of flash drought has raised growing concerns [54]. Given the rapid onset of flash drought, the absence of effective early warning systems can lead to substantial losses in various sectors, including agricultural production, water resources management, regional water security, and the national economy in Xinjiang. This concern is especially pronounced during critical crop growth stages, such as seed germination, pollination, and irrigation. Flash drought has the potential to significantly impact crop growth and development, ultimately affecting crop yields and agricultural productivity [55].

The role of human activities in influencing traditional drought must not be underestimated. For instance, the cultivation of cotton crops, which are highly water-intensive and require substantial irrigation during the summer, is prevalent in the marginal regions surrounding the Tarim Basin in southern Xinjiang, where traditional drought predominantly occurs (Figure 10h). The expansion of cotton cultivation has led to increased groundwater usage for irrigation in the oasis [56]. Consequently, ecological water resources have diminished, inhibiting plant growth within the desert ecosystem. This, in turn, has led to the emergence of extensive barren land, intensifying soil moisture evaporation and

plant transpiration during the summer [57]. The limited capacity to replenish soil moisture renders the region more susceptible to traditional drought.

Furthermore, it is imperative to address the risk of both flash drought and traditional drought in fragile ecological regions. When comparing 1995 to 2019, there has been an increase in population density, an expansion of human activities, and a significant rise in arable land and urban development (Figure 3). These anthropogenic activities have exacerbated the urban heat island effect and contributed to warming trends [58]. This nonlinear intensification of the relationship between human activities and climate further amplifies the risk of both flash drought and traditional drought.

The study reveals a significant climate shift in Xinjiang during the mid to late 1980s, transitioning from dry to wet climate [59]. However, a recent investigation by Yao et al. (2021a, 2021b) has raised a concerning trend. Despite fluctuations in temperature and precipitation at elevated levels since the early 21st century, the rate of increase has decelerated, suggesting that Xinjiang's climate is displaying indications of transitioning from a warm and wet pattern to a warm and dry one [60,61]. This short-term warming and humidification issue may be closely linked to a marked decrease in the severity of flash drought and a simultaneous increase in their frequency observed from 1990 to 2021. Extreme or anomalous increases in precipitation can alleviate the severity of drought and halt the development of prolonged, severe drought events, potentially transforming them into shorter-term drought events. It is essential to recognize that while the trend of traditional drought in recent years, as shown in Figure 12, may not be statistically significant, there remains an overall upward trend in drought occurrences. Due to the low precipitation base in the northwest arid region, primarily in Xinjiang, the absolute increase in precipitation is not significant compared with the increase in single precipitation intensity or strong precipitation. This change cannot fundamentally change the desert landscape pattern and the state of aridity and water scarcity in the northwest [62].

It is important to address that this study has certain limitations. The use of ERA5-Land may not fully capture soil moisture conditions, and the inclusion of desert areas may contribute to unexplained drought occurrences in these regions. In the future research, it is advisable to consider the integration of in situ soil moisture data to analyze the differential impacts of flash drought and traditional drought on vegetation.

## 5. Conclusions

This study utilized soil moisture data from ERA5-Land to establish a robust framework for differentiating flash drought from traditional drought based on historical events. The Geodetector was employed to investigate the driving factors influencing the spatial heterogeneity of these two drought types. The key findings of this study are summarized as follows:

- Distinct Drought Patterns: The research underscores distinctive drought patterns in Xinjiang. Southern Xinjiang exhibited a propensity for traditional drought, driven by its exceedingly hot and dry climate. Conversely, northern Xinjiang was more susceptible to flash drought with its relatively wetter conditions.
- Increasing Frequency and Severity: The trend of frequency and severity varies with changes in the sliding window. The frequency of both flash drought and traditional drought exhibited an upward trend since the 1990s. However, this trend is statistically significant only in the case of flash drought. Additionally, the severity of both flash drought and traditional drought displayed a noteworthy and substantial increase within sliding windows ranging from 46 to 62 years, passing the significance test ($\alpha \leq 0.05$).
- Precipitation as a Primary Driver: Precipitation emerges as the primary driver influencing the spatial distribution of both flash drought and traditional drought. All factors exerted a more potent explanatory force for flash drought compared to traditional drought. Moreover, environmental variables (vegetation type, soil type, normalized difference vegetation index, and topography type) exhibited a more substantial impact

on flash drought, while human activities exerted a more significant influence on traditional drought. Furthermore, interactions involving human activities had the potential to significantly amplify the explanatory power of the spatial heterogeneity for both flash drought and traditional drought.

- Elevated Risk in Vulnerable Regions: Significant variations in risk of flash drought unfolded across various ecological regions, with the highest risk occurring in mildly fragile ecological zones. Conversely, the risk of traditional drought displayed less variability across ecological regions. Moreover, a discernible uptick in flash drought risk in severely fragile zones emerged when comparing the results from 1995 to 2019.

These findings contribute to a deeper comprehension of flash drought and traditional drought dynamics within arid and semi-arid regions, ultimately underpinning the enhancement of drought monitoring and early warning systems.

**Author Contributions:** Writing—original draft, software, validation, investigation, resources, data curation, J.Z.; Conceptualization, methodology, funding acquisition, writing—review and editing, M.Z.; Visualization, data curation, writing—review and editing, J.Y.; Software, validation, Y.Y.; Formal analysis, project administration, R.Y. All authors have read and agreed to the published version of the manuscript.

**Funding:** This research was funded by the National Natural Science Foundation of China Youth Fund Project, grant number [12101598]; BR Program Youth Project of Chinese Academy of Sciences, grant number [2021000088]; and the Third Xinjiang Comprehensive Scientific Research Project on Comprehensive Evaluation and Sustainable Utilization of Land Resources in the Tuha Basin Subproject, grant number [2022xjkk110502].

**Data Availability Statement:** The raw data supporting the conclusion of this article will be made available by the authors, without undue reservation.

**Conflicts of Interest:** The authors declare that they have no conflict of interest or personal relationships that might affect the work reported in this study.

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
