# Peer review of "Comparison of Flash Drought and Traditional Drought on Characteristics and Driving Forces in Xinjiang"

_remotesensing, doi:10.3390/rs15194758_

Round 1

Reviewer 1 Report

I am very glad to review the paper in greater depth because the subject is an interesting and well-done study for arid and semi-arid regions, and the findings are of considerable interest. This study constructed a new model to identify flash droughts and traditional droughts in Xinjiang based on historical drought events, further clarified the spatial distribution and trend changes, and quantified the contribution of spatial heterogeneity factors. However, the paper still needs a few minor revisions before its publication in Remote Sensing. I have included a line-by-line review of the text below.

Specific comments:

Abstract:

L11-12: “flash and traditional droughts” should change to “flash drought and traditional droughts”. The terms “flash and traditional droughts” should be written separately.

L12: “The geographic detector” should change to  “The Geodetector”

L64: I thought this paragraph is about the purpose and main content of the study. Perhaps separate these into a new paragraph.

Methods:

L124: The calculation of PET and VPD should be described in detail.

L136-138: The method makes sense. This ensures the rate of onset or intensification if the drought drops from a wet state to a drought state in a short time. This is the difference between flash droughts and traditional droughts.

Results:

Fig.4: Delete the picture about “Snowfall”

L304: Can you explain in more detail the two-factor nonlinear enhancement relationship and the two-factor enhancement relationship?

Conclusion:

L385: “from 1960 to 202” should change to “from 1960 to 2021”

Author Response

We greatly appreciate your encouraging comments and suggestions, which have significantly enhanced the language quality of the manuscript. 

Please refer to the attached file for more details.

Reviewer 2 Report

Flash drought and traditional drought are compared in the study. For this aim, the frequency and severity of flash and traditional droughts are assessed. The subject is very important and the study is valuable in terms of climate change & drought severity but the flowchart of the methodology is missing. Some suggestions and comments to the authors are presented below:

1. A basic flowchart of the suggested methodology should be presented in the paper. Thus, the readers can easily follow the application procedures.

2. Literature part is looking weak. Give new and last updated examples from literature about “drought characteristics (duration, severity, and intensity)” as

doi.org/10.1080/02626667.2021.1934473

doi.org/10.3390/app10030913

3. The performance metrics are missing in the paper. Some metrics can be calculated to evaluate the application results.

4. Some statistical properties as coefficient of variation, confidence intervals, distribution characteristics, min and median, etc. of used data should be given in a table.

5. Which method is used for the spatial analysis of maps? It should be explained in detail.

6. Is there any limitation or classification for the application? Or is it only valid for Xinjiang?

7. Are there any criteria for the calibration and validation procedures (especially for precipitation) in the application?

Check the tenses of the sentences. There are present, present perfect and past tenses in a paragraph. See the paragraph in the Abstract.

There are some crucial errors.

Keywords should be ordered A to Z. Two more keywords as “frequency” and “severity” can be added to keywords.

Use passive sentences. Check the sentences started by “we”. See the Abstract …

One sentence can’t be a paragraph. See the lines 150-151 …

Author Response

Thank you very much for your thorough review of the paper. Your constructive comments and valuable suggestions have helped us make significant improvements to the manuscript both in terms of scientific content and language quality. We have added the flowchart of the methodology in the revision according to your comments. The following are the point-to-point responses to your comments.

Please refer to the attached file for more details.

Reviewer 3 Report

The manuscript addresses important and recently increasingly prominent topic of flash drought occurrence. Authors used relatively standard methods, developed particularly for analyses of flash droughts in the continental US, such as 20-/40-percentile thresholds of soil moisture and applied them to a so-far less studied region.

I believe the manuscript contains results worth publishing, however I have some comments and questions for authors I deem important to be addressed:

-The calculation of soil moisture percentiles needs to be clarified. Was the annual variation considered? Percentiles should be usually calculated for each grid and day-of-year separately (within some time-window for more robust results), otherwise they can be misleading.

-Part of the study area is extremely dry, desert area. Is there really a reason to study flash droughts particularly there, while there is probably no agriculture or forests, which could be affected in such a rapid event? Deserts are usually affected particularly by long-term droughts, which can lead to underground water resources depletion.

Also, did you check the soil moisture variability in the driest parts of the region? Sometimes, it is so low within such areas, that it does not make statistical sense to calculate percentiles.

-Is there a specific reason to call ERA5 “Eurocentric”? Yes, it has been produced by ECMWF, but as far as I now it has been developed fully as a Global product, similarly to CFSR which is also not viewed as “US-centric”.

-Fig.1 – Caption should include description of individual parts of the Figure and explain all abbreviations.

-There are many spelling mistakes within the manuscript. Please, check it carefully again or send it to a proofreading service.

-I don’t particularly like, that authors call soil moisture a “driver” of long-term drought in the conclusions. Low soil moisture is the effect not the cause of a drought. Please, revise the discussion and conclusions and make sure to differentiate what could be an actual driver of the drought and what is just accompanying phenomenon. Results of a statistical analysis are not everything, they need to be interpreted carefully and correctly.

Author Response

Thank you very much for your thorough review of the manuscript. Your constructive comments and valuable suggestions have helped us significantly improve the manuscript. The following are the point-to-point responses to your comments.

Please refer the attached file for more details.

Reviewer 4 Report

The article was well written and very interesting. The abstract, introduction, results and discussion were completely appropriate. If the following two positions are corrected, there is no problem for the article to be published in my opinion.

1.       The method of references at the end of the article is not in accordance with the rules of the journal. This should be corrected.

2.       Conclusion should be given case by case (bullets or numbers) with more details. These results are a little short for your study and need more interpretation.

Author Response

We appreciate your encouraging comments and suggestions.

Please refer the attached file for more details.

Round 2

Reviewer 2 Report

I suggest accepting the manuscript. The authors carefully revised the paper by answering each comment from the first round.

Reviewer 3 Report

Authors responded sufficiently to comments and concerns in the first review round and made appropriate adjustments in the manuscript. I have no further comments and I recommend the manuscript to be published.